# Association between Complex *ACTN3* and *ACE* Gene Polymorphisms and Elite Endurance Sports in Koreans: A Case–Control Study

**DOI:** 10.3390/genes15091110

**Published:** 2024-08-23

**Authors:** Ji Heon Chae, Seon-Ho Eom, Sang-Ki Lee, Joo-Ha Jung, Chul-Hyun Kim

**Affiliations:** 1Department of Sports Medicine, Soonchunhyang University, Asan 31538, Republic of Korea; kb062344@gmail.com (J.H.C.); ush1995@sch.ac.kr (S.-H.E.); 2Department of Physical Education, Korea National Sports University, Seoul 05541, Republic of Korea; lsk6607@knsu.ac.kr; 3Center for Sport Science in Chungnam, Asan 31580, Republic of Korea

**Keywords:** *ACTN3*, *ACE*, polymorphism, genotype, alleles, endurance, elite athletes, Koreans

## Abstract

*ACTN3 R577X* and *ACE I/D* polymorphisms are associated with endurance exercise ability. This case–control study explored the association of *ACTN3* and *ACE* gene polymorphisms with elite pure endurance in Korean athletes, hypothesizing that individuals with both *ACTN3 XX* and *ACE II* genotypes would exhibit superior endurance. We recruited 934 elite athletes (713 males, 221 females) and selected 45 pure endurance athletes (36 males, 9 females) requiring “≥90% aerobic energy metabolism during sports events”, in addition to 679 healthy non-athlete Koreans (361 males, 318 females) as controls. Genomic DNA was extracted and genotyped for *ACTN3 R577X* and *ACE I/D* polymorphisms. *ACE ID* (*p* = 0.090) and *ACTN3 RX*+*XX* (*p* = 0.029) genotype distributions were significantly different between the two groups. Complex *ACTN3-ACE* genotypes also exhibited significant differences (*p* = 0.014), with dominant complex genotypes positively affecting endurance (*p* = 0.039). The presence of *RX*+*II* or *XX*+*II* was associated with a 1.763-fold higher likelihood of possessing a superior endurance capacity than that seen in healthy controls (90% CI = 1.037–3.089). Our findings propose an association of combined *ACTN3 RX*+*XX* and *ACE II* genotypes with enhanced endurance performance in elite Korean athletes. While causality remains to be confirmed, our study highlights the potential of *ACTN3-ACE* polymorphisms in predicting elite endurance.

## 1. Introduction

Recent research has highlighted DNA polymorphisms contributing to prowess in certain types of sports [1]. Polymorphisms with a frequency of ≥1% can be classified as genetic markers of power, muscle strength, and endurance [2]. According to studies by Ahmetov et al. [1] and Semenova et al. [3], the number of known markers positively correlated with specific abilities in elite athletes has increased from 11 to 41 between 2014 and 2023.

Over 70% of the athletic ability in elite athletes is considered to have a genetic basis, with the rest being derived from the interactions of genetics with exogenous factors [4,5,6]. Endurance, which is the ability to repeatedly perform a physical movement under minimal load over a long duration, is determined by cardiovascular factors that affect oxygen supply to skeletal muscle components and shows considerable variation among individuals [7,8,9,10,11]. Endurance can thus be defined as a phenotype arising from athletic ability-related genotypes. Because it is a complex phenotype, thousands of DNA polymorphisms need to be considered, and a consistent, strong level of evidence is required to prove their contribution [1,12,13,14,15]. To date, *ACTN3 R577X* and *ACE I/D* have shown stronger associations with endurance exercise ability than other candidate genes and thus have been extensively researched [16,17,18].

The *ACTN3* gene, which codes for α-actinin-3, a structural protein that forms part of the cytoskeleton at the Z-line, providing spatial and structural stability, is located on chromosome 11 (11q13.2) [19,20]. The *R577X* null polymorphism exhibits a C to T change in the 1747th base on exon 16, resulting in an early stop codon in place of arginine (Arg; R) at position 577 (*R577X*) [21,22,23,24]. Individuals homozygous for the *X* allele show a complete absence of α-actinin-3, which is expressed only in type-II (fast twitch) muscle fibers, those responsible for rapid force production [25,26,27,28,29]. Deficiency or low expression of this protein in type-II fibers does not cause muscle disease [30]. Rather, as a compensation, structural protein α-actinin-2, which can be found in all types of fatigue-tolerable muscle fibers (such as skeletal, cardiac, and smooth muscle fibers), is upregulated. Therefore, α-actinin-2 is compensated for α-actinin-3 on the *X* allele, which functions as fatigue-tolerable continuous contractile muscle fiber [28,29].

The *ACTN3 R577X* polymorphism, which can determine α-actinin-3 protein expression, has direct functional effects that arise from the variation in structure and fiber phenotype of skeletal muscles [26,31]. The deficit or low expression of the α-actinin-3 protein was found to improve endurance in Australian elite athletes [31]. Moreover, a study of professional soccer players in Brazil found that athletes with the *XX* genotype, which results in complete α-actinin-3 protein deficiency, exhibited greater aerobic ability than athletes with the *RR* genotype possessing the full amount of α-actinin-3 protein [32]. Further, a number of studies have indicated that *ACTN3* deficiency and low expression are significant predictors of endurance, affecting muscle fiber composition and metabolic efficiency [28,31,32].

The *ACE* gene, another endurance-associated candidate gene, codes for the angiotensin I-converting enzyme that catalyzes angiotensin I (C62H89N17O14) degradation to its inactivated state and converts it to the physiologically active peptide angiotensin II (C50H71N13O12) within the renin-angiotensin system [33,34]. ACE activity is closely regulated by genotype-dependent expression as well as by endogenous inhibitory and secretory mechanisms [35]. Approximately 50% of the interindividual variance in ACE levels is determined by the *ACE* gene [36]. It is located on chromosome 17 (17q23.3) and is present as one of two alleles—the *I* (insertion) allele and *D* (deletion) allele—depending on the insertion of a 287bp Alu repetitive sequence in intron 16 [37].

In many studies, the *II* genotype and *I* allele have been significantly associated with elite endurance athletes [38,39,40,41]. Compared with the *D* allele, the *I* allele shows approximately one-third as much ACE activity [42,43]. According to Zhang et al., individuals with the *II* genotype, who carry the *ACE I* allele and thus have lower serum and tissue ACE activity, exhibited a higher proportion of type-I fibers and a lower proportion of type-II fibers in the left vastus lateralis compared to those with the *DD* genotype [44]. Another study found that individuals carrying the *ACE I* allele (*II*+*ID*) who underwent endurance training had over three-fold the volume density of subsarcolemmal and intramyocellular lipids compared to those with the *DD* genotype [45]. Consequently, individuals with lower ACE activity have relatively lower blood levels of angiotensin II [43,45], resulting in vasodilation and reduced blood pressure [46,47,48,49,50]. This was found to be advantageous for endurance by improving substrate delivery and skeletal muscle efficiency, thus preserving energy stores [14,41,51,52]. Conversely, research in African and Caucasian individuals revealed that, at 50% or 80% maximal oxygen consumption, the blood lactate concentration tended to be lower in those with the *DD* genotype than in those with the *II* genotype, suggesting that the *DD* genotype could be more efficient for endurance [21,53,54,55,56].

As of June 2023, approximately 110–120 studies have been conducted on endurance and the *ACTN3 R577X* or *ACE I/D* polymorphism, but the results have been inconsistent [17,57,58,59,60,61,62]. These discrepancies regarding gene polymorphisms and endurance appear to be due to differences in participant recruitment and group categorization between studies, in addition to differences between ethnicities affecting the recorded genetic characteristics [42,58,63,64,65]. As has been described in numerous review articles, ethnic diversity within study cohorts can result in considerable genetic background variation, which translates into differences in gene expression and function across ethnic groups [3,21]. It is therefore inappropriate to indiscriminately extrapolate the significance of genetic characteristics across different ethnic groups. Recently, an increasing number of studies have not only focused on single gene polymorphisms for the above-described two genes but have also combined multiple additional genes to corroborate the genetic basis of endurance [42,66]. However, there is a paucity of correlation studies that assess the relationships between these genes and endurance, particularly among Koreans. Such studies would highlight the phenotypes when combined polymorphisms of *ACTN3* and *ACE* are present.

In the present study, we investigated the association of complex genotypes of *ACTN3 R577X* and *ACE I/D* polymorphisms with elite pure endurance in Korean athletes. We hypothesized that individuals of Korean descent who possessed both the *ACTN3 XX* and the *ACE II* genotypes would exhibit superior levels of elite endurance. We used a strict criterion of “≥90% aerobic energy metabolism during sporting events”, which refers to the contribution of aerobically produced ATP utilized during sports [67]. The combination of genotypes with polymorphisms in multiple genes was defined as a complex genotype (CG) and classified as either endurance dominant (EDCG), neutral (ENCG), or recessive (ERCG), based on a previous study [68]. Using these classifications, we investigated the relationships of *ACTN3 R577X* and *ACE I/D* polymorphisms with endurance in elite Korean athletes.

## 2. Materials and Methods

### 2.1. Participants

In this case–control study, we recruited 713 male and 221 female elite athletes, with at least 5 years of experience representing their nation and placements within the top three of a national or international competition. Of the 934 athletes, 36 male and 9 female athletes in a discipline requiring “≥90% aerobic energy metabolism during sporting events” (marathon, 10,000 m run, 5000 m run, and 10~20 km race walk), as suggested by Powers [67], were included in the elite pure endurance athletes (EPEA) group, while the other 889 participants were excluded. The control group consisted of 679 non-athlete Korean apparently healthy individuals (361 male and 318 female participants) who participated in medical checkups. All study participants were healthy and without genetic disease. The general characteristics of the two groups are shown in Table 1. This study was approved by the institutional review board of Soonchunhyang University (number: 202211-BC-124-01) according to the Declaration of Helsinki, and written consent was obtained from all subjects before participation.

### 2.2. Blood Sampling and Genomic DNA Extraction

The participants provided 3-cc blood samples after 12 h of fasting, and the samples were stored as whole blood in anticoagulant-coated ethylenediaminetetraacetic acid (EDTA) tubes in a refrigerator at 4 °C. Genomic DNA was isolated using the Puregene^®^ Blood Kit (Cat. 158023; QIAGEN, Hilden, North Rhine-Westphalia, Germany).

### 2.3. Genotyping

To analyze the *ACTN3 R577X* polymorphism (rs1815739) in extracted gDNA, we used the previously validated MGB TaqMan^®^ SNP Genotyping Assay (Cat. 4351379, C_590093_1_; Applied Biosystems, Waltham, MA, USA), and to analyze the *ACE I/D* polymorphism (rs1799752), we used three primers (Cat. 4304970; Applied Biosystems) and two probes (Cat. 4316034; Applied Biosystems). The base sequences used in this analysis are shown in Table 2 [69].

The materials for gene amplification of *ACTN3* (MGB TaqMan^®^ SNP Genotyping Assay (20×), 10–20 ng of gDNA) and *ACE* (10 pmol of each primer, 150 nM *I* allele specific probe, 75 nM *D* allele specific probe, and 10–50 ng of gDNA) were added to the TaqMan^®^ Genotyping Master Mix (2X) (Cat. 4371353; Applied Biosystems) and nuclease-free water to make a reaction mix with a final volume of 10 μL. Target gene amplification was performed using the CFX Connect Real-Time PCR Detection System (Cat. 1855201; Bio-Rad, Hercules, CA, USA). The cycling profile consisted of two steps, with 10 min of enzyme activation at 95 °C, followed by 40 cycles of denaturation for 15 s at 95 °C and annealing and extension for 1 min at 60 °C. The allelic discrimination plot for identifying genotypes was plotted automatically using CFX Maestro Software 2.3 for Windows PC (Cat. 12013758; Bio-Rad).

### 2.4. Statistical Analysis

The frequency of each genotype and allele for the *ACTN3 R577X* and *ACE I/D* polymorphisms was calculated. The Hardy–Weinberg equilibrium (HWE) in each group was estimated using a chi-squared test (*p* < 0.05). We compared the frequencies of genotypes (2 × 3) and alleles (2 × 2) for the *ACTN3 R577X* and *ACE I/D* polymorphisms between the EPEA and control groups. For *ACTN3 R577X* polymorphism genotypes, we used the chi-squared test to compare their frequency (2 × 2) with an *X* allele, which are expected to be beneficial for endurance since they show phenotypical characteristics of type-I (slow twitch) muscle fibers owing to compensation by α-actinin-2 [31,70,71].

Additionally, to compare the differences in frequencies of complex *ACTN3-ACE* polymorphisms between the groups, we categorized genotype combinations as EDCGs, ENCGs, or ERCGs (2 × 3), and used the chi-squared test. If chi-squared tests indicated significant differences between the groups for a given genotype, we performed multiple comparisons (2 × 2) and adjusted the resulting *p*-value using Bonferroni correction.

The trend for genotype combinations that are beneficial for endurance (2 × 6) was analyzed using linear-by-linear association analysis. We also performed binary logistic regression analysis to obtain odds ratios (ORs) with a 90% confidence interval (CI) for the relative frequency of genotypes that are beneficial for pure elite endurance in the control group. The significance level for all comparisons was *p* < 0.10 [72,73,74]. Statistical analyses were performed using SPSS (version 26.0; IBM Corp., Armonk, NY, USA).

For the post hoc power analysis, a generic F test was performed using the G*power 3.1.9.7 software program, and the parameters α = 0.1, numerator df = 45-1, and denominator df = 679-1 were entered in the section “post hoc: compute power test”. The results of the power analysis were β = 0.0981 and (1-β) = 0.9019.

## 3. Results

### 3.1. Distribution of ACTN3 R577X and ACE I/D Polymorphisms among the Non-Athlete Korean Population and EPEA

In our study, the frequencies of *ACTN3 R577X* and *ACE I/D* polymorphism genotypes and alleles in the EPEA and control groups were all shown via HWE to be representative of the genotype distribution in each group (*p* > 0.05). As shown in Figure 1, the frequencies of the *ACE I* allele and *D* allele in the EPEA group were 53.3% and 46.7%, respectively, while those in the control group were 57.5% and 42.5%, respectively. For the distribution of *ACE I/D*, the respective frequencies of the *II*, *ID*, and *DD* genotypes were 35.6%, 35.6%, and 28.9% in the EPEA group versus 32.0%, 51.1%, and 16.9% in the control group, respectively. A significant difference in the distribution of the three *ACE I/D* (*Χ*^2^ = 5.615, df = 2, *p* = 0.060) and *ID* (*Χ*^2^ = 4.081, df = 1, *p* = 0.090) genotypes was noted between the groups. *ACTN3 R577X R* allele and *X* allele frequencies in the EPEA group were 47.8% and 52.2%, respectively, while *ACE I* allele and *D* allele frequencies in the control group were 55.2% and 44.8%, respectively. Regarding the distribution of *ACTN3 R577X* genotypes, the respective frequencies of *RR*, *RX*, and *XX* genotypes were 15.6%, 64.4%, and 20.0% in the EPEA group versus 29.5%, 51.5%, and 19.0% in the control group, respectively. Despite no significant differences in the distribution of the three *ACTN3 R577X* genotypes between the groups (*Χ*^2^ = 4.215, df = 2, *p* = 0.122), statistically significant differences were observed in the distributions of *RX*+*XX* alleles, which are associated with skeletal muscle characteristics beneficial for endurance, as well as in the distribution of the genotype (*RR*) without the *X* allele (*Χ*^2^ = 3.994, df = 1, *p* = 0.029).

### 3.2. Complex ACTN3-ACE Polymorphisms and Pure Elite Endurance in Koreans

We divided complex *ACTN3-ACE* polymorphisms into EDCGs, ENCGs, and ERCGs, in order of the positive effects predicted for pure elite endurance, to construct groups of complex genotypes (Table 3). We observed significant differences in the distribution of the three complex genotype groups between the EPEA and control groups (*Χ*^2^ = 8.460, df = 2, *p* = 0.014). We observed significant differences in distribution between the groups when using either the dominant complex genotypes or the recessive complex genotypes as a reference (EDCGs, *Χ*^2^ = 5.835, df = 1, *p* = 0.039; ERCGs, *Χ*^2^ = 5.641, df = 1, *p* = 0.021).

### 3.3. Genetic Relationships between Complex ACTN3-ACE Genotypes and Elite Pure Endurance in Koreans

To investigate the trends in relationships between complex *ACTN3-ACE* genotypes and pure endurance, we constructed complex genotype groups on the basis of exercise physiology, as shown in Table 4. When we analyzed trends for the frequencies of these complex genotype groups in the EPEA and control groups using linear-by-linear association analysis, we observed statistically significant differences (*Χ*^2^ = 6.079, df = 1, *p* = 0.007). Additionally, when we estimated ORs for each complex genotype group using binary logistic regression analysis, the relative likelihood of endurance dominance was significant at 1.763 for the *RX*+*XX*/*II* genotype (90% CI = 1.037–3.089).

## 4. Discussion

In the present study, we demonstrate that EPEAs are more likely to carry the *X* allele but less likely to have the *ID* heterozygous genotype than controls. Additionally, based on the following genotypes, when Koreans possess either the *RX*+*II* or *XX*+*II* complex genotype, they are 1.763 times more likely to be elite endurance athletes.

Various studies have demonstrated that the *RX* and *XX* genotypes are associated with endurance phenotypes [28,31,32,68,75,76,77]. Israeli long-distance athletes showed higher frequencies of the *XX* genotype and *X* allele than the controls [75]. Similarly, elite Australian endurance athletes showed a higher rate of the *XX* genotype than the controls [31]. In line with our findings, previous studies on elite Chinese endurance athletes [78], international-level Japanese long-distance runners [79], and elite Russian endurance athletes [62] showed an increased ratio of the *RX* genotype and not the *XX* genotype when compared with controls. These patterns suggest that while the *XX* genotype, which is considered to help endurance performance, is important for endurance athletes, one potentially decisive aspect of athletic performance required for competition is end spurts. Thus, since α-actinin-2 alone may be insufficient for adequate end spurts, the *RX* genotype may also be important [80].

Differences in skeletal muscle fiber types due to the presence of *ACTN3 R577X* are considered to be related to the characteristics required for outstanding endurance. MacArthur et al. reported that *ACTN3* knockout mice do not show significant changes in skeletal muscle fiber type distribution but do exhibit a decrease in anaerobic metabolism-related enzymatic activity within type-II fibers and an increase in aerobic metabolism-related activity, suggesting that α-actinin-3 deficiency could shift type-II fiber metabolism toward efficient aerobic metabolism pathways, ultimately resulting in improved endurance [28,76]. Hogarth et al. demonstrated that the level of α-actinin-3 protein expression was 50% lower for individuals with the *RX* genotype than for those with the *RR* genotype, while α-actinin-2 protein expression was increased [81]. This implies that because α-actinin-3 protein expression in the *RX* genotype was lower than that of the *RR* genotype, compensatory expression of α-actinin-2 protein, along with increased mitochondria count and capillary density, induced type-II fibers to exhibit type-I fiber-like characteristics [29,82,83,84,85,86,87]. Similar findings have been confirmed in human studies [32,70,88]. Since α-actinin-3 protein is only expressed in type-II fibers, α-actinin-3 deficiency in the *XX* genotype can be regulated by α-actinin-2 protein [26,89]. Owing to the decrease in type-II fibers, this could be related to the proportion of muscle fiber types, which significantly influences athletic ability [70].

According to studies on *ACE I/D* polymorphisms and endurance phenotypes, the frequency of the *I* allele is significantly higher in elite endurance athletes, including elite British mountaineers (≥7000 m) [90], elite white Australian rowers [38], elite white British runners (≥5000 m) [91], as well as in Spanish marathon and cross-country runners [40], when compared to that among controls. Conversely, in elite Iranian endurance cyclists, the frequency of the *D* allele was 27.02% higher than that of the *I* allele [92]. In elite Greek runners [60], the ratio of the *DD* genotype was higher than that of the *I* allele (*DD*: 42.86% and *I*: 17.85%). Moreover, several studies reported no relationships between *ACE I/D* polymorphisms and endurance [53,61,93], including studies of black Kenyan long-distance athletes (3000 m to a marathon) [63] and Australian aerobic sports athletes [94]. Our finding of a weak relationship between the *ACE ID* genotype and endurance ability (*p* = 0.090) is similar to the result of another East Asian study of elite Japanese long-distance athletes [95], in which the athletes showed a lower frequency of the *ID* genotype than that among controls. Although we were unable to calculate running speed as performed in the Japanese study, the current results could help explain the mixed phenomena observed in the aforementioned studies. Although limitations exist in understanding the effect that single gene polymorphisms have on endurance [96,97,98], we hypothesize that *ACE I/D* polymorphisms are only beneficial for endurance in homozygotes.

Although the precise mechanisms explaining the relationship between *ACE I/D* polymorphisms and elite endurance have not yet been established, Williams et al. found that the delta efficiency of muscle contraction increased considerably only for the *II* genotype [99]. Endurance ability due to the *I* allele is predicted to be caused by an independent local muscle mechanism [91,100], and low ACE activity in the *II* genotype increases the local nitrogen oxide concentration in skeletal muscle [99,101]. Japanese individuals with the *ACE II* genotype showed a much higher ratio of type-I fibers than those with the *DD* genotype (59.4% vs. 19.6%), whereas the ratio of type-IIb fibers was much lower (25.5% vs. 50.9%) [44]. This study was conducted with non-athletes rather than elite endurance athletes, and no significant change in muscle capillaries was observed depending on the *ACE I/D* genotype [44]. However, in a later muscle biopsy study on endurance, an increase in muscle capillaries was observed in *I* allele carriers, and, when compared with a group with the *II* genotype, a training group with the *DD* genotype showed a lower glycogen concentration in the vastus lateralis to replenish carbohydrate stores in muscle fibers [102].

Here, we demonstrated that *ACE* and *ACTN3* genotypes are related to endurance phenotypes in elite Korean endurance athletes. When we analyzed trends for each complex genotype in the control and EPEA groups using linear-by-linear association analysis, we found that participants with the *ACTN3 X* allele and homozygous *ACE* genotype tended to exhibit superior endurance than those without (*p* = 0.007). The *RX*+*XX*/*II* complex genotypes, which include the *ACE II* genotype, showed 11.2% higher frequency in the EPEA group than in the control group, revealing the highest tendency for endurance ability. Thus, in Koreans, the *RX*+*XX*/*II* complex genotype can predict endurance ability, and these findings are consistent with those of a previous study that used less strict criteria than our study, while also showing that these two genes are related to endurance ability in Koreans [68]. Our findings indicate that the combination of *ACTN3 *RX*+/italic>XX* and *ACE II* genotypes is associated with enhanced endurance performance in Korean elite athletes. This partially supports the initial hypothesis that Koreans with both the *ACTN3 XX* and *ACE II* genotypes would exhibit superior elite endurance. While the presence of the *ACTN3 XX* genotype alone was not significantly correlated with superior endurance, the combined effect of the *ACTN3 X* allele and *ACE II* genotype played a crucial role in enhancing endurance capacity in Koreans. A considerable number of other cohort studies on *ACTN3 R577X* and *ACE I/D* polymorphisms have repeatedly shown that *RX*/*XX* and *II*/*ID* genotypes are related to endurance [11,42,103,104], in line with our findings. However, several studies were unable to identify any effect of *ACTN3 R577X* and *ACE I/D* polymorphisms on endurance [58,105]. These discrepancies could stem from differences in participant ethnicity, training environment, or study design [6,7].

Although *ACTN3* and *ACE* are strongly associated with athletic ability, the findings regarding their roles in endurance ability have been inconsistent, and their effects on endurance in elite athletes could differ greatly depending on the interactions with other factors, including other polymorphisms [42]. The interactions between skeletal muscle contraction and the regulation of blood supply in endurance exercise are complex and can differ situationally [106,107,108]. However, muscle contraction typically occurs first, regulating blood flow [87,109] and affecting endurance ability. Thus, endurance ability can be improved by facilitating blood flow via skeletal muscle contraction, supplying nutrients required for muscle metabolism [110] and removing metabolic byproducts [7,111]. Considering that blood flow regulation by muscle fibers could be beneficial for endurance, we aimed to group complex genotypes and analyze their relationships with endurance. We found that the *ACE DD* genotype could be considerably suppressed by the *ACTN3*
*RX*+*XX* genotypes, whereas the *ACE II* genotype was suppressed less than the *ACE DD* genotype, presenting a new perspective. We surmised that, in the event of muscle contraction where the type-II fibers show type-I-like (*RX* genotype) or type-I (*XX* genotype) characteristics, this could be beneficial for endurance ability when blood flow is unimpeded owing to low *ACE* activity but not in the case of high *ACE* activity. This could help explain the intricacy of complex genotypes in humans.

This study has few limitations that should be considered. First, there is an imbalance in the number of cases between the two groups. While the total sample size provides sufficient power for the study, the smaller number of cases in the EPEA group compared to that in the control group may have influenced the results, despite our best efforts to conduct robust statistical analyses [3,112]. As of 2023, the total number of specialist athletes (college and general) in Korean athletics disciplines was 951, and the number of elite national athletes in disciplines requiring “≥90% aerobic energy metabolism during sporting events” is far lower. It is encouraging to note that by exploring the *ACTN3-ACE* complex polymorphism in a less-studied population, we were able to discern potential trends and effects in Koreans, thereby providing direction for future research. To overcome the limitations of this study and enhance its reliability, future studies should measure and collect data on additional variables that can represent endurance in the groups as well as involve larger-scale research with more participants.

Second, we cannot exclude the effects of participant ethnicity, training environment, or other factors on the physical phenotype of endurance [6]. Although Koreans show very high homogeneity compared with other ethnicities to the extent that they are called a monoethnic population [113], other findings might be observed in different ethnicities, including Caucasian or African people. Even within a given ethnic group, endurance performance can also be affected by the training environment within each athlete’s team or other psychological factors [7]. Our findings should be used as basic data to help identify talent, cultivate new athletes, and develop genotype-specific individualized training programs for Koreans.

Third, this study had a case–control design. Although we were able to demonstrate relationships between genes and endurance, we were unable to prove causality [3,6]. The new perspectives that we have presented thus remain hypothetical. To verify hypotheses about the relationship between muscle fiber contraction and blood supply, studies that include molecular biology experiments, such as measuring differences in *ACE* activity or blood flow to muscles following type-II fiber contraction in individuals with the *RX* or *XX* genotype, are warranted. Furthermore, preclinical experiments involving genetic manipulation to confirm complex genotypes and investigate related mechanisms are also required. In summary, our study highlights the importance of investigating and understanding the role of *ACTN3-ACE* complex polymorphisms in endurance.

Although we only identified certain possibilities regarding endurance and polymorphisms, this study provides valuable insights that can be utilized by field staff, such as coaches and trainers, to identify athletic talents. Therefore, *ACTN3-ACE* complex polymorphisms could be used as a tool to predict elite endurance in Koreans. This can facilitate the identification of athletes and their efficient development through individualized athletic training or injury-prevention training based on the individual’s natural characteristics. We hope that the current findings will not only help identify sports-related talent and foster new athletes but will also serve as foundational data for physiological and biochemical insights into the effects of gene polymorphisms on endurance exercise ability.

## Figures and Tables

**Figure 1 genes-15-01110-f001:**
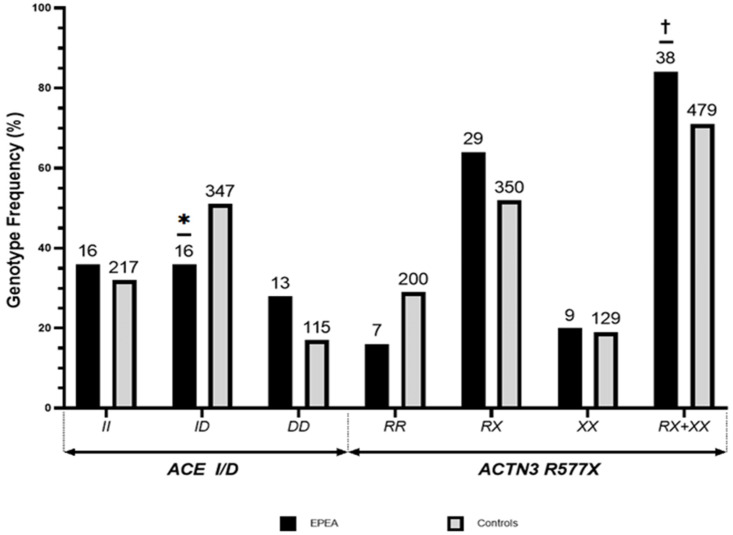
Distribution of *ACTN3 R577X* and *ACE I/D* genotypes in elite Korean endurance athletes. Values are shown as percentages (%). The number of cases is indicated above the bars. *, the mean *ID* genotype frequency in the EPEA group was significantly lower than that in the control group (*p* < 0.10 by Bonferroni correction). †, the mean *RX*+*XX* genotype frequency in the EPEA group was significantly higher than that in the control group (*p* < 0.10 via chi-squared test).

**Table 1 genes-15-01110-t001:** Demographics of the Korean EPEA and control groups (*n* = 724).

		EPEA	Controls
		(*n* = 45)	(*n* = 679)
Sex			
	Male	36 (80.0%)	361 (53.2%)
	Female	9 (20.0%)	318 (46.8%)

Age		20.6 ± 4.4	32.6 ± 4.8

Sport events			
	Marathon	15 (33.3%)	
	10,000 m run	10 (22.2%)	
	5000 m run	18 (40.0%)	
	10~20-kmW	2 (4.4%)	

Values are shown as numbers (%) and as means ± standard deviations. Note: EPEA, elite pure endurance athletes group; kmW, race walk.

**Table 2 genes-15-01110-t002:** Sequences used for *ACE I/D* polymorphism analysis.

Material	Designation	Sequence
Primers	*ACE* 111 ^a^	CCC-ATC-CTT-TCT-CCC-ATT-TCT-C
	*ACE* 112 ^b^	AGC-TGG-AAT-AAA-ATT-GGC-GAA-AC
	*ACE* 113 ^a^	CCT-CCC-AAA-GTG-CTG-GGA-TTA
Probes	*I* allele-specific ^c^	AGG-CGT-GAT-ACA-GTC-A
	*D* allele-specific ^d^	TGC-TGC-CTA-TAC-AGT-CA

Note: ^a^, forward 5′-3′; ^b^, reverse 5′-3′; ^c^, forward VIC 5′-3′MGB; ^d^, forward FAM 5′-3′MGB.

**Table 3 genes-15-01110-t003:** Complex genotype combinations of the *ACTN3-ACE* genes.

	Complex Genotypes		
	EDCGs	ENCGs	ERCGs
	** *RX* ** **+*XX*/*II*+*DD***	** *RX* ** **+*XX*/*ID*** ***RR*/*II*+*DD***	** *RR* ** **/*ID***
EPEA	23 * (51.1%)	21 (46.7%)	1 ^†^ (2.2%)
Controls	227 (33.4%)	358 (52.7%)	94 (13.8%)

Values are shown as numbers (%). Note: EPEA, elite pure endurance athletes group; EDCGs, endurance-dominant complex genotypes; ENCGs, endurance-neutral complex genotypes; ERCGs, endurance-recessive complex genotypes; *, the mean EDCG frequency in the EPEA group was significantly higher than that in the control group (*p* < 0.10 via Bonferroni correction); †, the mean ERCG frequency in the EPEA group was significantly lower than that in the control group (*p* < 0.10 via Bonferroni correction). Complex genotype distribution (2 × 3 contingency table), *Χ*^2^ = 8.460, *p* < 0.10. EDCGs vs. remainder group (2 × 2 contingency table), *Χ*^2^ = 5.835 [df = 1], *p* < 0.10 via Bonferroni correction. ENCGs vs. remainder group (2 × 2 contingency table), *Χ*^2^ = 0.621 [df = 1], *p* > 0.10 via Bonferroni correction. ERCGs vs. remainder group (2 × 2 contingency table), *Χ*^2^ = 5.641 [df = 1], *p* < 0.10 via Bonferroni correction.

**Table 4 genes-15-01110-t004:** Complex genotype combinations of the *ACTN3* and *ACE* genes based on exercise physiology.

Complex Genotypes		EPEA (*n* = 45)	Controls (*n* = 679)	*p* (Linear by Linear Association)	EPEA vs. Controls
					ORs (90% CI)
*RX*+*XX*	*II* (*n* = 165)	15 (33.3%)	150 (22.1%)	0.007	1.763 (1.037–3.089)
	*DD* (*n* = 85)	8 (17.8%)	77 (11.3%)		1.690 (0.863–3.221)
	*ID* (*n* = 267)	15 (33.3%)	252 (37.1%)		0.847 (0.504–1.468)
*RR*	*II* (*n* = 69)	1 (2.2%)	68 (10.0%)		0.264 (0.051–1.235)
	*DD* (*n* = 43)	5 (11.1%)	38 (5.6%)		2.109 (0.945–4.742)
	*ID* (*n* = 95)	1 (2.2%)	94 (13.8%)		0.183 (0.036–0.847)

Values are shown as numbers (%). Complex genotype distribution (2 × 6 contingency table), linear-by-linear association *Χ*^2^ = 6.079 [df = 1], *p* < 0.10. Odds ratio (ORs) of complex genotypes in endurance, *RX*+*XX*/*II* odds ratio mean: 1.763, 90% CI: 1.037–3.089. Note: *RX*+*XX*/*II*, *RX*+*II*, and *XX*+*II* combinations; reference, remainder genotypes; CI, confidence interval.

## Data Availability

The datasets generated and/or analyzed during the study will be available from the corresponding author upon reasonable request.

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
