# Peer review of "Association between Complex ACTN3 and ACE Gene Polymorphisms and Elite Endurance Sports in Koreans: A Case–Control Study"

_genes, 2024, doi:10.3390/genes15091110_

Round 1

Reviewer 1 Report

Comments and Suggestions for Authors

General Comments:

The authors of the present manuscript present an investigation that investigates an interesting examination of the interplay between two very relevant genes associated with athletic performance: ACTN3 and ACE I/D. However, there minor structural issues, and moreover glaring methodological flaws that absolutely need to be addressed.  

Given the glaring issues observed in the methods and preceding lack of associated rationale in the introduction, I do not believe it is appropriate to evaluate the current discussion. Namely, any interpretations made from these findings might be erroneous. Nonetheless, any grammatical issues

Methods:

A power analysis should be used to rationalize the number of participants you included in the present investigation. You discuss being potentially underpowered in the discussion but this is asinine without a better idea of what an appropriate subject pool would be. It is somewhat hard to believe that you would be underpowered with the number of participants recruited, which is additionally why a metric of effect sizes should be included to demonstrate any magnitude of changes. Lastly, I am entirely confused as to why a p value of >0.1. is used as the significance threshold? Especially in the exercise science realm, I have never seen this and it ultimately appears as though you are reaching to make statements about your analyses to be statistically significant.

Results:

The combination of alleles/ genotypes such as RX+XX seems to be arbitrary. You mention it as a predictive indicator for endurance performance on page 6, lines 197-198, but this is not substantiated in the introduction nor is there a citation. I see this again with the “complex genotypes” re-groupings. This makes sense to some degree, but what rationale is there for some of the heterogeneous groupings to represent a vague category like endurance “neutral” complex genotypes.

Specific Comments:

Page 1, Lines 12-12: There is no purpose in your abstract or question that rationalizes it.

Page 1, Line 14: A sex/gender breakdown would be beneficial here

Page 2, Line 53: the alpha-actinin gene is a structural protein that facilitates high-force conduction/transfer through the tissue. Your description gives the impression that it is a contractile protein like actin or myosin, which is not the case from a traditional interpretation.

Page 2, Lines 62-63. A transition sentence would facilitate the connection between the end of this paragraph and the beginning of the next.

Page 2, Lines 75-77: Can you provide more rationale as to how angiotensin II facilitates this change in substrate delivery? Physiologically, does it cause renal vasoconstriction and peripheral vasodilation for increased blood flow? Be more descriptive.

Page 3, Line 86: This is the first time you address ethnicity. More detail is needed here as to how this factor contributes to variability.

Page 3, Lines 89-91: This final sentence is difficult to read and should be rewritten. Also, there is description in the discussion but none in the introduction that rationalize as to why Korean individuals are of particular interest to investigate.

Page 3, Line 96: is “rate of oxygen consumption” just VO2? This language is different to many studies in the literature.

Page 3, Line 99: “Hope” is a fairly non-scientific term. I suggest changing this to something more appropriate and highly recommend inserting both purpose and hypothesis statements.

Page 3, Line 107 & 112: You should generally elude to the type of athletes (i.e. events, sports, etc.) represented, even if there is a breakdown in Table 1. Furthermore, what was the criteria for a non-athlete? There is potentially a large difference between a recreational marathon runner and a sedentary individual.

Page 5, Table 2: Why are the sequences for ACTN3 not listed?

Page 6, Lines 181-183: This sentence is confusing. I suggest you reword it.

Page 7, Figure 1: Why are standard deviations or errors not included in this figure? This might help explain the non-significant findings.

Page 7-9, Tables 3 & 4: The table should not be broken from one page to another, so please be mindful of this moving forward.

Reviewer 2 Report

Comments and Suggestions for Authors

Journal: Genes
Manuscript ID: genes-3116335
Type of manuscript: Article
Title: The genetic association between the complex polymorphism of ACTN3 and ACE genes and elite endurance sports in Koreans: A case-control study

 In the work presented to me for review entitled "The genetic association between the complex polymorphism of ACTN3 and ACE genes and elite endurance sports in Koreans: A case-control study" compared the genetic profiles of ACTN3 and ACE genes of the elite Korean endurance athletes with high rate of oxygen consumption with non-athlete Korean individuals. 

My comments:

1)     Authors should check the abstract with the rest of the paper. The objectives of the work should be uniform and the hypothesis should be described in the conclusions.

Abstract: " This study aimed to investigate the relationship between ACTN3 R577X and ACE I/D polymorphisms and endurance in elite Korean athletes using a strict criterion of “≥90% rate of oxygen consumption” to assess endurance.”

 "The hypothesis was that Koreans with both the ACTN3 XX and the ACE II genotypes would exhibit superior elite endurance."

The conclusions lack an answer to the hypothesis ?

 This text should be more suitable for a summary

"We hope that our study findings will not only help to identify sports-related talent and foster new athletes, but also to serve as foundational data with implications for physiological and biochemical perspectives on the effects of gene polymorphisms on endurance exercise ability. "

 2)     The title of Table 1 is too general and should be clarified. Explanations should be written in a smaller font below the Table or Figure.

3)     Is the institutional review board a bioethics committee working in accordance with the Helsinki Declaration?

4)     Check that all abbreviations are explained.

Please re-read your work to improve it.

Comments on the Quality of English Language

Minor linguistic inaccuracies and lack of commas in appropriate places.

Round 2

Reviewer 1 Report

Comments and Suggestions for Authors

Counter-response: The authors have done a sufficient job covering many of my concerns, however, there are still some glaring issues. Some of these could be more easily amended, but I am uncertain if your efforts can substantiate some of the choices made at this point in time.

1. We acknowledge the importance of power analysis in study design to justify the appropriate sample size. As anticipated, the overall sample size does have sufficient power. However, our discussion about potential underpowering stems from the imbalance in sample sizes between the two groups.

Counter-response: Could you then provide a post-hoc power analysis in the statistical analysis section to demonstrate that you are adequately powered?

2. We understand the concern regarding the use of a p-value threshold of >0.1 instead of the conventional 0.05. Traditionally, the 0.05 threshold is widely accepted for indicating statistical significance in exercise science. It is important to differentiate between the statistical and practical significance of findings. While a p-value <0.05 is conventionally used to denote statistical significance, a p-value <0.1 can still provide valuable insights, particularly in exploratory studies. Our study is exploratory in nature, focusing on a less-studied population. Using a p-value <0.1 allows us to identify potential trends and effects that merit further investigation, despite not meeting the stricter p <0.05 threshold.

Several studies across various fields, including exercise science, have used this threshold to indicate trends that require additional validation. We believe that using a p <0.1 threshold allows for the exploration of ACTN3 R577X and ACE I/D polymorphisms in a less-studied population such as the Korean population and can identify subtle yet potential biologically significant effects.

Counter-response: Perhaps I am ignorant of the use of p>0.1 as a critical threshold, so please provide evidence to substantiate this point. Without such data, I am hard pressed to say these statistics are rigorous.

General Comments 2: Results: The combination of alleles/ genotypes such as RX+XX seems to be arbitrary. You mention it as a predictive indicator for endurance performance on page 6, lines 197-198, but this is not substantiated in the introduction nor is there a citation. I see this again with the “complex genotypes” re-groupings. This makes sense to some degree, but what rationale is there for some of the heterogeneous groupings to represent a vague category like endurance “neutral” complex genotypes.

G. C. Response 2: We understand that the combination of RX+XX genotypes might appear arbitrary. However, as discussed on page 9 of the discussion section, various studies have demonstrated the association between RX+XX genotypes and endurance phenotypes. This is also thoroughly explained in the fifth line of the “2.4. Statistical Analysis” section. Although you mentioned that the introduction lacks supporting evidence, we have provided a detailed explanation from line 54 on page 2 until the end of that paragraph. To further address your concern, we have included a statement in the introduction that underscores the importance of ACTN3 deficiency (RX, XX genotypes) as a significant predictor of endurance performance.

Furthermore, our classification of "neutral" complex genotypes is a commonly used method in studies of candidate genes related to athletic performance. The "neutral" complex genotype category was created based on previous studies that showed no significant association between certain genotypes and physical performance. While the format may differ, conceptually, it encompasses the theory of the total genotype score. To clarify this, we have added citations to support the classification method.

Counter-response: I understand your reasoning for this grouping better now, which is much appreciated. However, it seems as though RX and XX are both being described as “deficient” in ACTN3, which is untrue to my understanding. The former should be better described with respect to its functional differences between RR and RX genotypes in the introduction to better substantiate your RX+XX stratification.

As you mentioned, the criteria for non-athletes can vary significantly among individuals, but this variation also reflects the characteristics of the Korean population. Furthermore, the endurance athlete group is defined by strict criteria, including a minimum of 5 years of athletic experience and a record of placing third or higher in international competitions. These rigorous conditions ensure a clear distinction between elite endurance athletes and non-athlete controls. The endurance capacity of elite athletes meeting these criteria is distinct from that of the general population. Our study aims to investigate the relationship between endurance capacity and genotypes; hence, we selected individuals with distinct endurance abilities by adhering to strict criteria for endurance athletes. The control group of non-athletes was selected based on health and the absence of genetic disorders, as described in the text.

Counter-response: I understand your reasoning for the seeming lack of detail on the control group. However, do you at least have some data on the activity habits of these individuals?

Author Response

Thank you.

Round 3

Reviewer 1 Report

Comments and Suggestions for Authors

I commend your team on the corrections made to the present version and for providing sufficient evidence to support your decisions otherwise.